# GROWING ACTION SPACES

## ABSTRACT

In complex tasks, such as those with large combinatorial action spaces, random exploration may be too inefficient to achieve meaningful learning progress. In this work, we use a curriculum of progressively growing action spaces to accelerate learning. We assume the environment is out of our control, but that the agent may set an internal curriculum by initially restricting its action space. Our approach uses off-policy reinforcement learning to estimate optimal value functions for multiple action spaces simultaneously and efficiently transfers data, value estimates, and state representations from restricted action spaces to the full task. We show the efficacy of our approach in proof-of-concept control tasks and on challenging large-scale StarCraft micromanagement tasks with large, multi-agent action spaces.

## 1 INTRODUCTION

The value of curricula has been well established in machine learning, reinforcement learning, and in biological systems. When a desired behaviour is sufficiently complex, or the environment too unforgiving, it can be intractable to learn the behaviour from scratch through random exploration. Instead, by "starting small" (Elman, 1993), an agent can build skills, representations, and a dataset of meaningful experiences that allow it to accelerate its learning. Such curricula can drastically improve sample efficiency (Bengio et al., 2009).

Typically, curriculum learning uses a progression of tasks or environments. Simple tasks that provide meaningful feedback to random agents are used first, and some schedule is used to introduce more challenging tasks later during training (Graves et al., 2017). However, in many contexts neither the agent nor experimenter has such unimpeded control over the environment. In this work, we instead make use of curricula that are internal to the agent, simplifying the exploration problem without changing the environment. In particular, we grow the size of the action space of reinforcement learning agents over the course of training.

At the beginning of training, our agents use a severely restricted action space. This helps exploration by guiding the agent towards rewards and meaningful experiences, and provides low variance updates during learning. The action space is then grown progressively. Eventually, using the most unrestricted action space, the agents are able to find superior policies. Each action space is a strict superset of the more restricted ones. This paradigm requires some domain knowledge to identify a suitable hierarchy of action spaces. However, such a hierarchy is often easy to find. Continuous action spaces can be discretised with increasing resolution. Similarly, curricula for coping with the large combinatorial action spaces induced by many agents can be obtained from the prior that nearby agents are more likely to need to coordinate. For example, in routing or traffic flow problems nearby agents or nodes may wish to adopt similar local policies to alleviate global congestion. Our method will be valuable when it is possible to identify a restricted action space in which random exploration leads to significantly more meaningful experiences than random exploration in the full action space.

We propose an approach that uses off-policy reinforcement learning to improve sample efficiency in this type of curriculum learning. Since data from exploration using a restricted action space is still valid in the Markov Decision Processes (MDPs) corresponding to the less restricted action spaces, we can learn value functions in the less restricted action space with 'off-action-space' data collected by exploring in the restricted action space. In our approach, we learn value functions corresponding to each level of restriction simultaneously. We can use the relationships of these value functions to

each other to accelerate learning further, by using value estimates themselves as initialisations or as bootstrap targets for the less restricted action spaces, as well as sharing learned state representations.

Empirically, we first demonstrate the efficacy of our approach in two simple control tasks, in which the resolution of discretised actions is progressively increased. We then tackle a more challenging set of problems with combinatorial action spaces, in the context of StarCraft micromanagement with large numbers of agents (50-100). Given the heuristic prior that nearby agents in a multiagent setting are likely to need to coordinate, we use hierarchical clustering to impose a restricted action space on the agents. Agents in a cluster are restricted to take the same action, but we progressively increase the number of groups that can act independently of one another over the course of training. Our method substantially improves sample efficiency on a number of tasks, outperforming learning any particular action space from scratch, a number of ablations, and an actor-critic baseline that learns a single value function for the behaviour policy, as in the work of Czarnecki et al. (2018). Code is available, but redacted here for anonymity.

## 2 RELATED WORK

Curriculum learning has a long history, appearing at least as early as the work of Selfridge et al. (1985) in reinforcement learning, and for the training of neural networks since Elman (1993). In supervised learning, one typically has control of the order in which data is presented to the learning algorithm. For learning with deep neural networks, Bengio et al. (2009) explored the use of curricula in computer vision and natural language processing. Many approaches use handcrafted schedules for task curricula, but others (Zaremba & Sutskever, 2014; Pentina et al., 2015; Graves et al., 2017) study diagnostics that can be used to automate the choice of task mixtures throughout training. In a self-supervised control setting, Murali et al. (2018) use sensitivity analysis to automatically define a curriculum over action dimensions and prioritise their search space.

In some reinforcement learning settings, it may also be possible to control the environment so as to induce a curriculum. With a resettable simulator, it is possible to use a sequence of progressively more challenging initial states (Asada et al., 1996; Florensa et al., 2017). With a procedurally generated task, it is often possible to automatically tune the difficulty of the environments (Tamar et al., 2016). Similar curricula also appear often in hierarchical reinforcement learning, where skills can be learned in comparatively easy settings and then composed in more complex ways later (Singh, 1992). Taylor et al. (2007) use more general inter-task mappings to transfer $Q$-values between tasks that do not share state and action spaces. In adversarial settings, one may also induce a curriculum through self-play (Tesauro, 1995; Sukhbaatar et al., 2017; Silver et al., 2017). In this case, the learning agents themselves define the changing part of the environment.

A less invasive manipulation of the environment involves altering the reward function. Such reward shaping allows learning policies in an easier MDP, which can then be transferred to the more difficult sparse-reward task (Colombetti & Dorigo, 1992; Ng et al., 1999). It is also possible to learn reward shaping on simple tasks and transfer it to harder tasks in a curriculum (Konidaris & Barto, 2006).

In contrast, learning with increasingly complex function approximators does not require any control of the environment. In reinforcement learning, this has often taken the form of adaptively growing the resolution of the state space considered by a piecewise constant discretised approximation (Moore, 1994; Munos & Moore, 2002; Whiteson et al., 2007). Stanley & Miikkulainen (2004) study continual complexification in the context of coevolution, growing the complexity of neural network architectures through the course of training. These works progressively increase the capabilities of the agent, but not with respect to its available actions.

In the context of planning on-line with a model, there are a number of approaches that use progressive widening to consider increasing large action spaces over the course of search (Chaslot et al., 2008), including in planning for continuous action spaces (Couëtoux et al., 2011). However, these methods cannot directly be applied to grow the action space in the model-free setting.

A recent related work tackling our domain is that of Czarnecki et al. (2018), who train mixtures of two policies with an actor-critic approach, learning a single value function for the current mixture of policies. The mixture contains a policy that may be harder to learn but has a higher performance ceiling, such as a policy with a larger action space as we consider in this work. The mixing coefficient is initialised to only support the simpler policy, and adapted via population based training

(Jaderberg et al., 2017). In contrast, we simultaneously learn a different value function for each policy, and exploit the properties of the optimal value functions to induce additional structure on our models. We further use these properties to construct a scheme for off-action-space learning which means our approach may be used in an off-policy setting. Empirically, in our settings, we find our approach to perform better and more consistently than an actor-critic algorithm modeled after Czarnecki et al. (2018), although we do not take on the significant additional computational requirements of population based training in any of our experiments.

A number of other works address the problem of generalisation and representation for value functions with large discrete action spaces, without explicitly addressing the resulting exploration problem (Dulac-Arnold et al., 2015; Pan et al., 2018). These approaches typically rely on action representations from prior knowledge. Such representations could be used in combination with our method to construct a hierarchy of action spaces with which to shape exploration.

## 3 BACKGROUND

We formalise our problem as a MDP, specified by a tuple $< \mathcal{S}, \mathcal{A}, P, r, \gamma >$. The set of possible states and actions are given by $\mathcal{S}$ and $\mathcal{A}$, $P$ is the transition function that specifies the environment dynamics, and $\gamma$ is a discount factor used to specify the discounted return $R = \sum_{t=0}^{T} \gamma^t r_t$ for an episode of length $T$. We wish our agent to maximise this return in expectation by learning a policy $\pi$ that maps states to actions. The state-action value function ($Q$-function) is given by $Q^\pi = \mathbb{E}_\pi[R|s, a]$. The optimal $Q$-function $Q^*$ satisfies the Bellman optimality equation:

$$Q^*(s, a) = \mathcal{T}Q^*(s, a) = \mathbb{E}[r(s, a) + \gamma \max_{a'} Q^*(s', a')]. \tag{1}$$

$Q$-learning (Watkins & Dayan, 1992) uses a sample-based approximation of the Bellman optimality operator $\mathcal{T}$ to iteratively improve an estimate of $Q^*$. $Q$-learning is an off-policy method, meaning that samples from any policy may be used to improve the value function estimate. We use this property to engage $Q$-learning for off-action-space learning, as described in the next section.

We also introduce some notation for restricted action spaces. In particular, for an MDP with unrestricted action space $\mathcal{A}$ we define a set of $N$ action spaces $\mathcal{A}_\ell, \ell \in \{0, \ldots, N-1\}$. Each action space is a subset of the next: $\mathcal{A}_0 \subset \mathcal{A}_1 \subset \ldots \subset \mathcal{A}_{N-1} \subseteq \mathcal{A}$. A policy restricted to actions $\mathcal{A}_\ell$ is denoted $\pi_\ell(a|s)$. The optimal policy in this restricted policy class is $\pi_\ell^*(a|s)$, and its corresponding action-value and value functions are $Q_\ell^*(s, a)$ and $V_\ell^*(s) = \max_a Q_\ell^*(s, a)$.

Additionally, we define a hierarchy of actions by identifying for every action $a \in \mathcal{A}_\ell, \ell > 0$ a parent action $\mathtt{parent}_\ell(a)$ in the space of $\mathcal{A}_{\ell-1}$. Since action spaces are subsets of larger action spaces, for all $a \in \mathcal{A}_{\ell-1}, \mathtt{parent}_\ell(a) = a$, i.e., one child of each action is itself. Simple pieces of domain knowledge are often sufficient to define these hierarchies. For example, a discretised continuous action can identify its nearest neighbour in $\mathcal{A}_{\ell-1}$ as a parent. In Section 5 we describe a possible hierarchy for multi-agent action spaces. One could also imagine using action-embeddings (Tennenholtz & Mannor, 2019) to learn such a hierarchy from data.

## 4 CURRICULUM LEARNING WITH GROWING ACTION SPACES

We build our approach to growing action spaces (GAS) on off-policy value-based reinforcement learning. $Q$-learning and its deep-learning adaptations have shown strong performance (Hessel et al., 2018), and admit a simple framework for off-policy learning.

### 4.1 OFF-ACTION-SPACE LEARNING

A value function for an action space $\mathcal{A}_\ell$ may be updated with transitions using actions drawn from its own action space, or any more restricted action spaces, if we use an off-policy learning algorithm. The restricted transitions simply form a subset of the data required to learn the value functions of the less restricted action spaces. To exploit this, we simultaneously learn an estimated optimal value function $\hat{Q}_\ell^*(s, a)$ for each action space $\mathcal{A}_\ell$, and use samples drawn from a behaviour policy based on a value function for low $\ell$ to directly train the higher $\ell$ value functions.

At the beginning of each episode, we sample $\ell$ according to some distribution. The experiences generated in that episode are used to update all of the $\hat{Q}^*_{\geq \ell}(s, a)$. This off-action-space learning is a type of off-policy learning that enables efficient exploration by restricting it to the low-$\ell$ regime. We sample at the beginning of the episode rather than at each timestep because, if the agent uses a high-$\ell$ action, it may enter a state that is inaccessible for a lower-$\ell$ policy, and we do not wish to force a low-$\ell$ value function to generalise to states that are only accessible at higher $\ell$.

Since data from a restricted action space only supports a subset of the state-action space relevant for the value functions of less restricted action spaces, we hope that a suitable function approximator still allows some generalisation to the unexplored parts of the less restricted state-action space.

## 4.2 VALUE ESTIMATES

Note that:
$$V_i^*(s) \leq V_j^*(s) \forall s \text{ if } i < j. \tag{2}$$
This is because each action space is a strict subset of the larger ones, so the agent can always in the worst case fall back to a policy using a more restricted action space.

This monotonicity intuitively recommends an iterative decomposition of the value estimates, in which $\hat{Q}^*_{\ell+1}(s, a)$ is estimated as a sum of $\hat{Q}^*_\ell(s, a)$ and some positive $\Delta_\ell(s, a)$. This is not immediately possible due to the mismatch in the support of each function. However, we can leverage a hierarchical structure in the action spaces when present, as described in Section 3. In this case we can use:
$$\hat{Q}^*_{\ell+1}(s, a) = \hat{Q}^*_\ell(s, \texttt{parent}_\ell(a)) + \Delta_\ell(s, a). \tag{3}$$
This is a task-specific upsampling of the lower-$\ell$ value function to intialise the next value function. Both $\hat{Q}^*_\ell(s, a)$ and $\Delta_\ell(s, a)$ are learned components. We could further regularise or restrict the functional form of $\Delta_\ell$ to ensure its positivity when $\texttt{parent}_\ell(a) = a$. However, we did not find this to be valuable in our experiments, and simply initialised $\Delta_\ell$ to be small.

The property (2) also implies a modified Bellman optimality equation:
$$Q_\ell^*(s, a) = \mathbb{E}[r(s, a) + \gamma \max_{i \leq \ell} \max_{a'} Q_i^*(s', a')] \tag{4}$$

The $\max_{i < \ell}$ are redundant in their role as conditions on the optimal value function $Q_\ell^*$. However, the Bellman optimality equation also gives us the form of a $Q$-learning update, where the term in the expectation on the RHS is used as an operator that iteratively improves an estimate of $Q^*$. When these estimates are inaccurate, the modified form of the Bellman equation may lead to different updates, allowing the solutions to higher $\ell$ to be bootstrapped from those at lower $\ell$.

We expect that policies with low $\ell$ are easier to learn, and that therefore the corresponding $\hat{Q}^*_\ell$ is higher value and more accurate earlier in training than those at high $\ell$. These high values could be picked up by the extra maximisation in the modified bootstrap, and thereby rapidly learned by the higher-$\ell$ value functions. Empirically however, we find that using this form for the target in our loss function performs no better than just maximising over $\hat{Q}^*_\ell(s', a')$. We discuss the choice of target and these results in more detail in Section 6.2.

## 4.3 REPRESENTATION

By sharing parameters between the function approximators of each $Q_\ell$, we can learn a joint state representation, which can then be iteratively decoded into estimates of $Q^*$ for each $\ell$. This shared embedding can be iteratively refined by, e.g., additional network layers for each $\hat{Q}^*_\ell$ to maintain flexibility along with transfer of useful representations. This simple approach has had great success in improving the efficiency of many multi-task solutions using deep learning (Ruder, 2017).

## 4.4 CURRICULUM SCHEDULING

We need to choose a schedule with which to increase the $\ell$ used by the behaviour policy over the course of training. Czarnecki et al. (2018) use population based training (Jaderberg et al., 2017) to choose a mixing parameter on the fly. However, this comes at significant computational cost, and

optimises greedily for immediate performance gains. We use a simple linear schedule on a mixing parameter $\alpha \in [0, N]$. Initially $\alpha = 0$ and we always choose $\ell = 0$. Later, we pick $\ell = \lfloor \alpha \rfloor$ with probability $\lceil \alpha \rceil - \alpha$ and $\ell = \lceil \alpha \rceil$ with probability $\alpha - \lfloor \alpha \rfloor$ (e.g. if $\alpha = 1.1$, we choose $\ell = 1$ with 90% chance and $\ell = 2$ with 10% chance). This worked well empirically with little effort to tune. Many other strategies exist for tuning a curriculum automatically (such as those explored by Graves et al. (2017)), and could be beneficial, at the cost of additional overhead and algorithmic complexity.

## 5 GROWING ACTION SPACES FOR MULTI-AGENT CONTROL

In cooperative multi-agent control, the full action space allows each of $N$ agents to take actions from a set $\mathcal{A}_{\text{agent}}$, resulting in an exponentially large action space of size $|\mathcal{A}_{\text{agent}}|^N$. Random exploration in this action space is highly unlikely to produce sensical behaviours, so growing the action space as we propose is particularly valuable in this setting. One approach would be to limit the actions available to each agent, as done in our discretised continuous control experiments (Section 6.1) and those of Czarnecki et al. (2018). However, the joint action space would still be exponential in $N$. We propose instead to use hierarchical clustering, and to assign the same action to nearby agents.

At the first level of the hierarchy, we treat the whole team as a single group, and all agents are constrained to take the same action. At the next level of the hierarchy, we split the agents into $k$ groups using an unsupervised clustering algorithm, allowing each group to act independently. At each further level, every group is split once again into $k$ smaller groups. In practice, we simply use $k$-means clustering based on the agent's spatial position, but this can be easily extended to more complex hierarchies using other clustering approaches.

To estimate the value function, we compute a state-value score $\hat{V}(s)$, and a group-action delta $\Delta_\ell(s, a_g, g)$ for each group $g$ at each level $\ell$. Then, we compute an estimated group-action value for each group, at each level, using a per-group form of (3): $\hat{Q}_{\ell+1}^*(s, a_g) = \hat{Q}_\ell^*(s, \texttt{parent}_k(a_g)) + \Delta_\ell(s, a_g, g)$. We use $\hat{Q}_{-1}^*(s, \cdot) = \hat{V}(s)$ to initialise the iterative computation, similarly to the dueling architecture of Wang et al. (2015). The estimated value of the parent action is the estimated value of the entire parent group all taking the same action as the child group. At each level $\ell$ we now have a set of group-action values.

In effect, a multi-agent value-learning problem still remains at each level $\ell$, but with a greatly reduced number of agents at low $\ell$. We could simply use independent $Q$-learning (Tan, 1993), but instead choose to estimate the joint-action value at each level as the mean of the group-action values for the groups at that $\ell$, as in the work of Sunehag et al. (2017). A less restrictive representation, such as that proposed by Rashid et al. (2018), could help, but we leave this direction to future work.

A potential problem is that the clustering changes for every state, which may interfere with generalisation as group-actions will not have consistent semantics. We address this in two ways. First, we include the clustering as part of the state, and the cluster centroids are re-initialised from the previous timestep for $t > 0$ to keep the cluster semantics approximately consistent. Second, we use a functional representation that produces group-action values that are broadly agnostic to the identifier of the group. In particular, we compute a spatially resolved embedding, and pool over the locations occupied by each group. See Figure 2 and Section 6.2 for more details.

## 6 EXPERIMENTS

We investigate two classes of problems that have a natural hierarchy in the action space. First, simple control problems where a coarse action discretisation can help accelerate exploration, and fine action discretisation allows for a more optimal policy. Second, the cooperative multi-agent setting, discussed in Section 5, using large-scale StarCraft micromanagement scenarios.

### 6.1 DISCRETISED CONTINUOUS CONTROL

As a proof-of-concept, we look at two simple examples: versions of the classic Acrobot and Mountain Car environments with discretised action spaces. Both tasks have a sparse reward of +1 when the goal is reached, and we make the exploration problem more challenging by terminating episodes

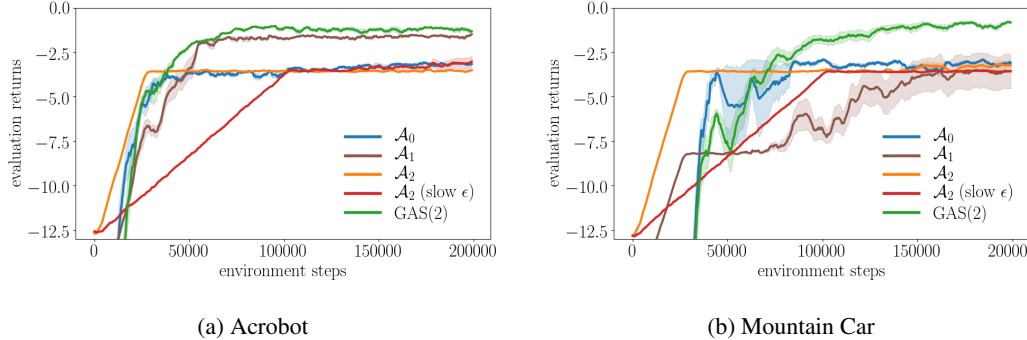

(a) Acrobot                                               (b) Mountain Car

Figure 1: Discretised continuous control with growing action spaces. We report the mean and standard error (over 10 random seeds) of the returns during training, with a moving average over the past 20 episodes. $\mathcal{A}_2$ (slow $\epsilon$) is an ablation of $\mathcal{A}_2$ that decays $\epsilon$ at a quarter the rate.

with a penalty of -1 if the goal is not reached within 500 timesteps. The normalised remaining time is concatenated to the state so it remains Markovian despite the time limit. There is a further actuation cost of $0.05\|a\|_2$. At $\mathcal{A}_0$, the actions apply a force of $+1$ and $-1$. At each subsequent $\mathcal{A}_{\ell>0}$, each action is split into two children, one that is the same as the parent action, and the other applying half the force. Thus, there are $2^\ell$ actions in $\mathcal{A}_\ell$.

The results of our experiments are shown in Figure 1. Training with the lower resolutions $\mathcal{A}_0$ and $\mathcal{A}_1$ from scratch converges to finding the goal, but incurs significant actuation costs. Training with $\mathcal{A}_2$ from scratch almost never finds the goal with $\epsilon$-greedy exploration. We also tried decaying the $\epsilon$ at a quarter of the rate ($\mathcal{A}_2$ slow $\epsilon$) without success. In these cases, the policy converges to the one that minimises actuation costs, never finding the goal. Training with a growing action space explores to find the goal early, and then uses this experience to transition smoothly into a solution that finds the goal but takes a slower route that minimises actuation costs while achieving the objective.

## 6.2 Combinatorial action spaces: StarCraft battles

### 6.2.1 Large-scale StarCraft Micromanagement

The real-time strategy game StarCraft and its sequel StarCraft II have emerged as popular platforms for benchmarking reinforcement learning algorithms (Synnaeve et al., 2016; Vinyals et al., 2017). Full game-play has been tackled by e.g. (Lee et al., 2018; Vinyals et al., 2019), while other works focus on sub-problems such as *micromanagement*, the low-level control of units engaged in a battle between two armies (e.g. (Usunier et al., 2016)). Efforts to approach the former problem have required some subset of human demonstrations, hierarchical methods, and massive compute scale, and so we focus on the latter as a more tractable benchmark to evaluate our methods.

Most previous work on RL benchmarking with StarCraft micromanagement is restricted to maximally 20-30 units (Samvelyan et al., 2019; Usunier et al., 2016). In our experiments we focus on much larger-scale micromanagement scenarios with 50-100 units on each side of the battle. To further increase the difficulty of these micromanagement scenarios, in our setting the starting locations of the armies are randomised, and the opponent is controlled by scripted logic that holds its position until any agent-controlled unit is in range, and then focus-fires on the closest enemy. This increases the exploration challenge, as our agents need to learn to find the enemy first, while they hold a strong defensive position. The action space for each unit permits an `attack-move` or `move` action in eight cardinal directions, as well as a `stop` action that causes the unit to passively hold its position.

In our experiments, we use $k = 2$ for $k$-means clustering and split down to at most four or eight groups. The maximum number of groups in an experiment with $\mathcal{A}_\ell$ is $2^\ell$. Although our approach is designed for off-policy learning, we follow the common practice of using $n$-step $Q$-learning to accelerate the propagation of values (Hessel et al., 2018). Our base algorithm uses the objective of $n$-step $Q$-learning from the work of Mnih et al. (2016), and collects data from multiple workers into a short queue similarly to Espeholt et al. (2018). Full details can be found in the Appendix.

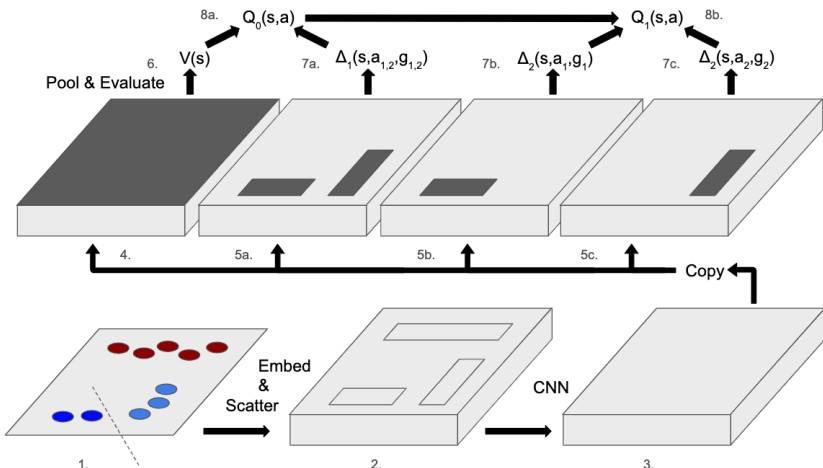

Figure 2: Architecture for GAS with hierarchical clustering. For clarity, only two levels of hierarchy are shown. The dark shaded regions identify the locations that are pooled over before state-value or group-action scores are computed.

### 6.2.2 MODEL ARCHITECTURE

We propose an architecture to efficiently represent the value functions of the action-space hierarchy. The overall structure is shown in Figure 2. We start with the state of the scenario (1). Ally units are blue and split into two groups. From the state, features are extracted from the units and map (see Appendix for full details). These features are concatenated with a one-hot representation of the unit's group (for allied agents), and are embedded with a small MLP. A 2-D grid of embeddings is constructed by adding up the unit embeddings for all units in each cell of the grid (2). The embeddings are passed through a residual CNN to produce a final embedding (3), which is copied several times and decoded as follows. First, a state-value branch computes a scalar value by taking a global mean pooling (4) and passing the result through a 2-layer MLP (6). Then, for each $\ell$, a masked mean-pooling is used to produce an embedding for each group at that $\mathcal{A}_\ell$ by masking out the positions in the spatial embedding where there are no units of that group (5a, 5b, 5c). A single evaluation MLP for each $\ell$ is used to decode this embedding into a group action-score (7a, 7b, 7c). This architecture allows a shared state representation to be efficiently decoded into value-function contributions for groups of any size, at any level of restriction in the action space.

We consider two approaches for combining these outputs. In our default approach, described in Section 5, each group's action-value is given by the sum of the state-value and group-action-scores for the group and its parents (8a, 8b). In 'SEP-Q', each group's action-value is simply given by the state-value added to the group-action score, i.e., $\hat{Q}_\ell^*(s, a_g) = \hat{V}(s) + \Delta_\ell(s, a_g, g)$. This is an ablation in which the action-value estimates for restricted action spaces do not initialise the action-value estimates of their child actions.

### 6.2.3 RESULTS AND DISCUSSION

Figure 3 presents the results of our method, as well as a number of baselines and ablations, on a variety of micromanagement tasks. Our method is labeled Growing Action Spaces GAS($\ell$), such that GAS(2) will grow from $\mathcal{A}_0$ to $\mathcal{A}_2$. Our primary baselines are policies trained with action spaces $\mathcal{A}_0$ or $\mathcal{A}_2$ from scratch. GAS(2) consistently outperforms both of these variants. Policies trained from scratch on $\mathcal{A}_2$ struggle with exploration, in particular in the harder scenarios where the opponent has a numbers advantage. Policies trained from scratch on $\mathcal{A}_0$ learn quickly, but plateau comparatively low, due to the limited ability of a single group to position effectively. GAS(2) benefits from the efficient exploration enabled by an intialisation at $\mathcal{A}_0$, and uses the data gathered under this policy to efficiently transfer to $\mathcal{A}_2$; enabling a higher asymptotic performance.

We also compare against a Mix&Match (MM) baseline using the actor-critic approach of Czarnecki et al. (2018), but adapted for our new multi-agent setting and supporting a third level in the mixture

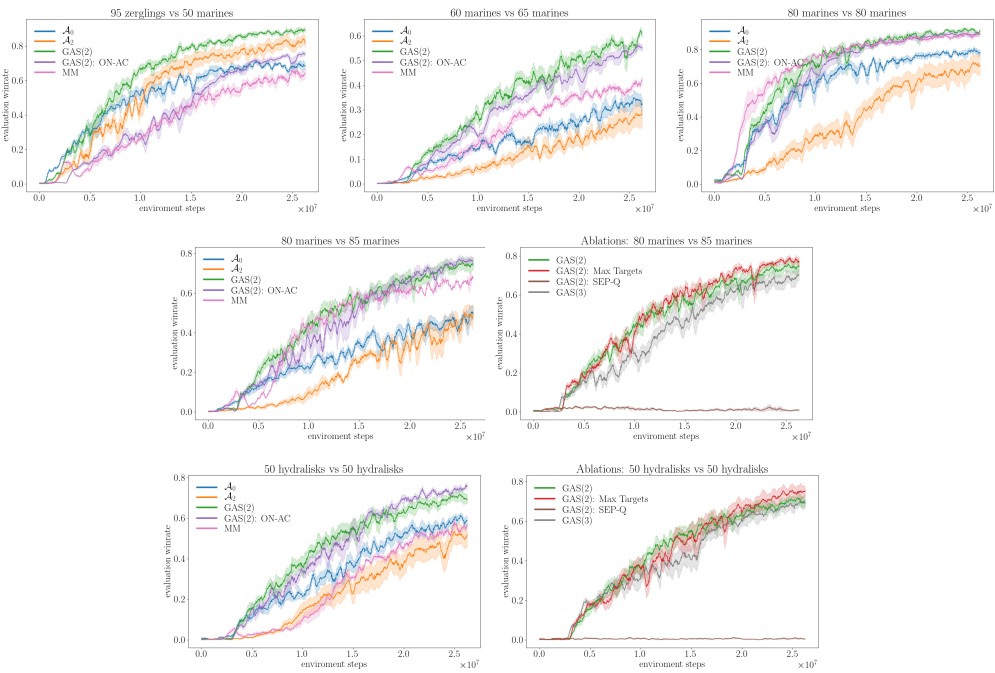

Figure 3: StarCraft micromanagement with growing action spaces. We report the mean and standard error (over 5 random seeds) of the evaluation winrate during training, with a moving average over the past 500 episodes.

of policies ($\mathcal{A}_0$, $\mathcal{A}_1$, $\mathcal{A}_2$). We tuned hyperparameters for all algorithms on the easiest, fastest-training scenario (80 marines vs. 80 marines). On this scenario, MM learns faster but plateaus at the same level as GAS(2). MM underperforms on all other scenarios to varying degrees. Learning separate value functions for each $\mathcal{A}_\ell$, as in our approach, appears to accelerate the transfer learning in the majority of settings. Another possible explanation is that MM may be more sensitive to hyperparameters. We do not use population based training to tune hyperparameters on the fly, which could otherwise help MM adapt to each scenario. However, GAS would presumably also benefit from population based training, at the cost of further computation and sample efficiency.

The policies learned by GAS exhibit good tactics. Control of separate groups is used to position our army so as to maximise the number of attacking units by forming a wall or a concave that surrounds the enemy, and by coordinating a simultaneous assault. Figure 5 in the Appendix shows some example learned policies. In scenarios where MM fails to learn well, it typically falls into a local minimum of attacking head-on.

In each scenario, we test an ablation GAS (2): ON-AC that does not use our off-action-space update, instead training each level of the $Q$-function only with data sampled at that level. This ablation performs somewhat worse on average, although the size of the impact varies in different scenarios. In some tasks, it is beneficial to accelerate learning for finer action spaces using data drawn from the off-action-space policy. In Appendix A.1.1, the same ablation shows significantly worse performance on the Mountain Car task and comparable performance on Acrobot.

We present a number of further ablations on two scenarios. The most striking failure is of the 'SEP-Q' variant which does not compose the value function as a sum of scores in the hierarchy. It is critical to ensure that values are well-initialised as we move to less restricted action spaces. In the discretised continuous control tasks, 'SEP-Q' also underperforms, although less dramatically.

The choice of target is less important: performing a max over coarser action spaces to construct the target as described in Section 4.2 does not improve learning speed as intended. One potential reason is that maximising over more potential targets increases the maximisation bias already present in

$Q$-learning (Hasselt, 2010). Additionally, we use an $n$-step objective which combines a partial on-policy return with the bootstrap target, which could reduce the relative impact of the choice of target.

Finally, we experiment with a higher $\ell$. Unfortunately, asymptotic performance is degraded slightly once we use $\mathcal{A}_3$ or higher. One potential reason is that it decreases the average group size, pushing against the limits of the spatial resolution that may be captured by our CNN architecture. Higher $\ell$ increases the amount of time that there are fewer units than groups, leaving certain groups empty and rendering our masked pooling operation degenerate. We do not see a fundamental limitation that should restrict the further growth of the action space, although we note that most hierarchical approaches in the literature avoid too many levels of depth. For example, Czarnecki et al. (2018) only mix between two sizes of action spaces rather than the three we progress through in the majority of our GAS experiments.

## 7 CONCLUSION

In this work, we presented an algorithm for growing action spaces with off-policy reinforcement learning to efficiently shape exploration. We learn value functions for all levels of a hierarchy of restricted action spaces simultaneously, and transfer data, value estimates, and representations from more restricted to less restricted action spaces. We also present a strategy for using this approach in cooperative multi-agent control. In discretised continuous control tasks and challenging multi-agent StarCraft micromanagement scenarios, we demonstrate empirically the effectiveness of our approach and the value of off-action-space learning. An interesting avenue for future work is to automatically identify how to restrict action spaces for efficient exploration, potentially through meta-optimisation. We also look to explore more complex and deeper hierarchies of action spaces.

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

## A    APPENDIX

### A.1    DISCRETISED CONTINUOUS CONTROL

#### A.1.1    ADDITIONAL ABLATIONS

Here, we present results on some additional ablations of GAS on the discretised continous control tasks. SEP-Q performs slightly worse on both tasks, a less dramatic failure than in the StarCraft experiments. These value functions are simpler, and it is easier to learn the new action space's value without relying so much on the previous one. ON-AC performs worse only on Mountain Car, suggesting once again that the significance of this component of the algorithm is somewhat problem-dependent. We also test a version that follows the intuition of the 'Match' objective of M&M more closely, adapted for the value-based setting: instead of using an adaptive initialisation of each level's $Q$-function as described in the main text, we use an L2 penalty to 'Match' the new level's value function to its parent action, which should have a similar effect. This variant performs similarly here (perhaps slightly worse in the more challenging Mountain Car).

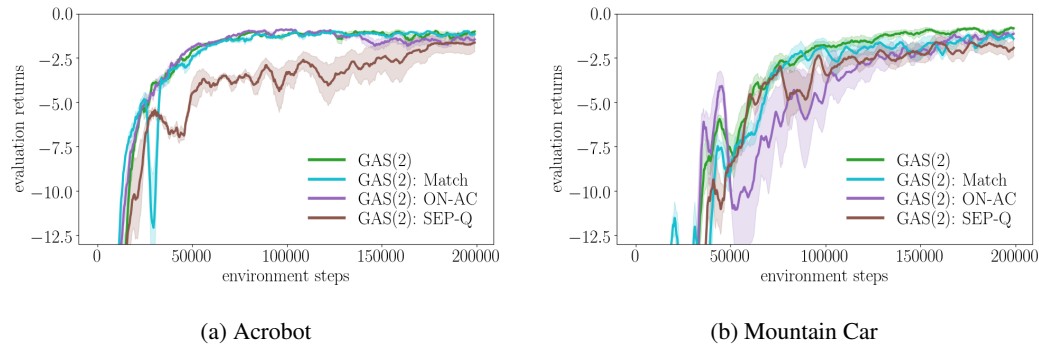

| (a) Acrobot | (b) Mountain Car |

Figure 4: Additional GAS ablations on the discretised continuous control tasks.

#### A.1.2    HYPERPARAMETERS

For our experiments in discretised continous control, we use a standard DQN trainer (Mnih et al., 2015) with the following parameters.

| Parameter | Value |
|---|---:|
| batch size | 128 |
| replay buffer size | 10000 |
| target update interval | 200 |
| $\epsilon$ initial | 1.0 |
| $\epsilon$ final | 0.1 |
| $\epsilon$ decay | 25000 env steps |
| $\ell$ lead-in | 25000 env steps |
| $\ell$ growth | 25000 env steps |
| env steps per model udpate | 4 |
| Adam learning rate | 5e-4 |
| Adam $\epsilon$ | 1e-4 |

For GAS experiments, we keep the mixing coefficient $\alpha = 0$ for 25000 environment steps, and then increase it linearly by 1 every 25000 steps until reaching the maximum value. We use $\gamma = 0.998$ for our Acrobot experiments, but reduce it to $\gamma = 0.99$ for Mountain Car to prevent diverging $Q$-values.

Our model consists of fully-connected ReLU layers, with 128 hidden units for the first and 64 hidden units for all subsequent layers. Two layers are applied as an encoder. Then, for each $\ell$ one layer is applied on the current embedding to produce a new embedding, and an evaluation layer on that embedding produces the $Q$-values for that level.

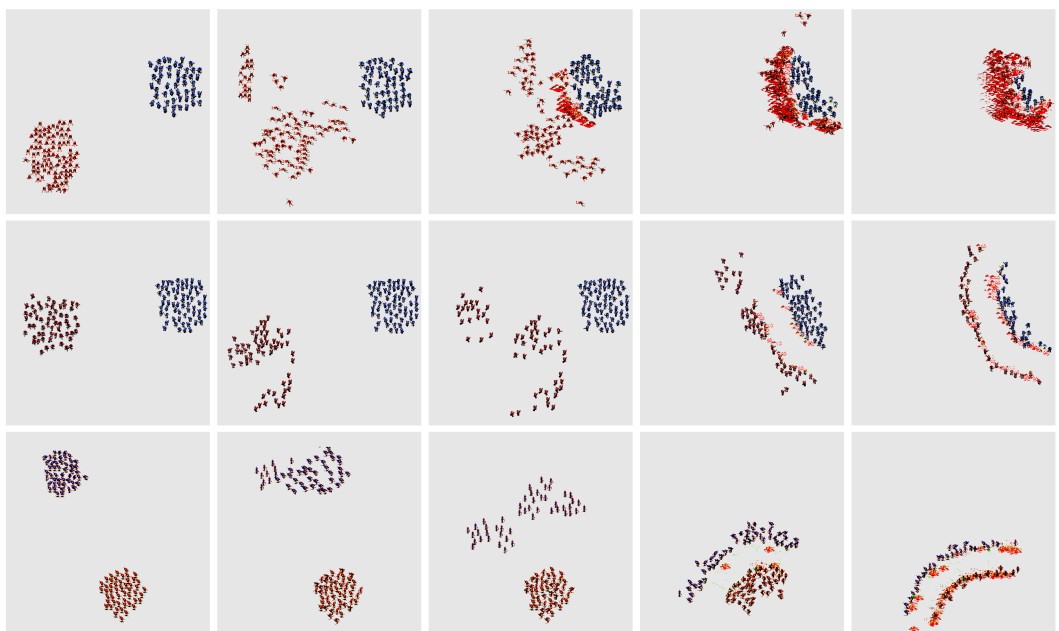

Figure 5: Final learned policies of StarCraft micromanagement unit control with growing action spaces. Scenarios shown from left to right at time 0, 3, 5, 10, 15 seconds. Top to bottom the scenarios are: 60 marines vs 65 marines, 50 hydralisks vs 50 hydralisks, 95 zerglings vs 50 marines. In these examples, the opponent is always on the right, and the agent controlled by model trained with GAS is on the left.

## A.2 STARCRAFT MICROMANAGEMENT SCENARIOS

### A.2.1 SCENARIOS AND LEARNED STRATEGIES

We explore five Starcraft micromanagement scenarios: 50 hydralisks vs 50 hydralisks, 80 marines vs 80 marines, 80 marines vs 85 marines, 60 marines vs 65 marines, 95 zerglings vs 50 marines. In these scenarios, our model controls the first set of units, and the opponent controls the second set.

The opponent is a scripted opponent that holds its location until an opposing unit is within range to attack. Then, the opponent will engage in an "attack-closest" behavior, as described in Usunier et al. (2016), where each unit individually targets the closest unit to it. Having the opponent remain stationary until engaged makes this a more difficult problem – the agent must find its opponent, and attack into a defensive position, which requires good positions prior to engagement.

As mentioned in section 6.2, all of our scenarios require control of a much larger number of units than previous work. The 50 hydralisks and 80v80 marines scenarios are both imbalanced as a result of attacking into a defensive position. The optimal strategy for 80 marines vs 85 marines and 60 vs 65 marines requires slightly more sophisticated unit positioning, and the 95 zerglings vs 50 marines scenario requires the most precise positioning. The agent can use the enemy's initial stationary positioning to its advantage by slightly surrounding the opponent in a concave, ensuring that the outermost units are in its attack range, but far enough away to be out of range of the center-most enemy units. Ideally, the timing of the groups in all scenarios should be coordinated such that all units get in range of the opponent at roughly the same point in time. Figure 5 shows how our model is able to exhibit this level of unit control.

### A.2.2 FEATURES

We use a standard features for the units and map, given by TorchcraftAI [1]

For each of the units, the following features are extracted:

---

[1]https://github.com/TorchCraft/TorchCraftAI

- Current x, y positions.
- Current x, y velocities.
- Current hitpoints
- Armor and damage values
- Armor and damage types
- Range versus both ground and air units
- Current weapon cooldown
- A few boolean flags on some miscellaneous unit attributes

Approximate normalization for each feature keep its value approximately between 0-1.

For the map, the following features are extracted for each tile in the map:

- a one-hot encoding of tile's the ground height (4 channels)
- boolean representing or not the given tile is walkable
- boolean representing or not the given tile is buildable
- and boolean representing or not the given tile is covered by fog of war.

The features form a $HxWx7$ tensor, where our map has height $H$ and width $W$.

### A.2.3 Environment details

We use a frame-skip of 25, approximately 1 second of real time, allowing for reasonably fine-grained control but without making the exploration and credit assignment problems too challenging.

We calculate at every timestep the difference in total health points (HP) and number of units for the enemy from the last step, normalised by the total starting HP and unit count. As a reward function, we use the normalised damage dealt, plus 4 times the normalised units killed, plus an additional reward of 8 for winning the scenario by killing all enemy units. This reward function is designed such that the agent gets some reward for doing damage and killing units, but the reward from doing damage will never be greater than from winning the scenario. Ties and timeouts are considered losses.

### A.3 Experimental details

### A.3.1 Model

As described in Section 6.2.2 a custom model architecture is used for Starcraft micromanagement. Each unit's feature vector is embedded to size 128 in step 2 of Figure 2. The grid where the unit features and map features are scattered onto is the size of the Starcraft map of the scenario in walktiles downsampled by a factor of 8. After being embedded, the unit features for ally and enemy units are concatenated with the downsampled map features and sent into a ResNet encoder with four residual blocks (stride 7 padding 3). The output is an embedding of size 64.

The decoder uses a mean pooling over the embedding cells as described in Section 6.2.2. Each evaluator is a 2-layer MLP with 64 hidden units and 17 outputs, one for each action. All layers are separated with ReLU nonlinearities.

### A.3.2 Training hyperparameters

We use 64 parallel actors to collect data in a short queue from which batches are removed when they are consumed by the learner. We use batches of 32 6-step segments for each update.

For the $Q$-learning experiments, we used the Adam optimizer with a learning rate of $2.5 \times 10^{-4}$ and $\epsilon = 1 \times 10^{-4}$. For the MM baseline experiments, we use a learning rate of $1 \times 10^{-4}$, entropy loss coefficient of $8 \times 10^{-3}$ and value loss coefficient 0.5. The learning rates and entropy loss coefficient were tuned by random search, training with $\mathcal{A}_0$ from scratch on the 80 marines vs 80

marines scenario with 10 configurations sampled from `log_uniform`$(-5, -3)$ for the learning rate and `log_uniform`$(-3, -1)$ for the entropy loss coefficient.

For $Q$-learning, we use an $\epsilon$-greedy exploration strategy , decaying $\epsilon$ linearly from 1.0 to 0.1 over the first 10000 model updates. We also use a target network that copies the behaviour model's parameters every 200 model updates.

We also use a linear schedule to grow the action-space. There is a lead in of 5000 model updates, during which the action-space is held constant at $\mathcal{A}_0$, to prevent the action space from growing when $\epsilon$ or the policy entropy is too high. The action-space is then grown linearly at a rate of 10000 model updates per level of restriction, so that after 10000 updates, we act entirely at $\mathcal{A}_1$ and after 20000, entirely at $\mathcal{A}_2$.

