# OpenReview forum: "Growing Action Spaces"
_ICLR.cc/2020/Conference — Reject_

### Official Review · AnonReviewer1 · 2019-10-22
**Official Blind Review #1**

**Rating:** 3

**Review:**

The paper presents a method of scaling up towards action spaces, that exhibit natural hierarchies (such as a controllable resolution of actions), throughout joint training of Q-functions over these. Authors notice, and exploit a few interesting properties, such as inequalities that emerge when action spaces form strict subsets that lead to nice parametrisation of policies in a differential way. The evaluation is performed in simple toy-ish tasks, and in micro-management problem in 5 scenarios in the game of SC2.

On a fundamental level, proposed method resembles that of Mix & Match, that authors discuss in the paper. In the M&M paper authors use the matching (distillation) of the policies, to ensure knowledge transfer, while in GAS, authors share information through said differential reparametrisation. Ablations provided imply that this part is indeed crucial (as with independent Qs, called "Sep-Q" learning flat-lines). The ablation testing the off-policy modification, seems a bit less conclusive, despite authors claiming that "This ablation performs slightly, or considerably, worse in each scenario. " We can see in Figure 3 that there were 5 experiments:
- in one [95z vs 50m] GAS works much better
- in two [80m v 80m, 80m v 85m] there seems to be no difference (in terms of longer term performance)
- in one [60m v 65m] GAS works slightly better
- in one [50h v 50h] GAS works slightly worse
This mixed bag of results would rather suggest that the offpolicy part is not the main contributing factor, and might require closer investigation to really understand which part of the system proposed brings the benefits. Could this ablation be also done on the toy-ish tasks from experiment 1? Given its simplicity it should be cheap enough to run these extra experiments (I am assuming the SC2 ones are quite expensive?)

Reviewer finds it hard to understand, given current description, how was M&M baseline adapted to the Q-learning setup? Was the distillation loss replaced with L2 one? Was the exact same architecture used for these experiments? How were the missing parent actions handled? Or did M&M experiments use an actor critic learning instead (which would make the comparison more about Q-learning vs Actor-Critic learning, than about methods of action space scaling). Methods section (mentioning entropy loss) looks like M&M was indeed trained with actor-critic, which would make the baseline hardly comparable. Adapting M&M strategy to Q-learning (or picking other baselines, that work on the same RL setup), in reviewer's opinion, is crucial for actual evaluation of author's contributions (given that this is the only baseline).

On a minor point - given how unique SC2 environment (and problem of unit micromanagement) is, would it be possible to provide baselines results also for the toy-ish experiment 1?

Overall, I recommend a weak rejection of the paper, and I am happy to revisit this evaluation given that authors address the above comments.

**Experience Assessment:**

I have published one or two papers in this area.

**Review Assessment: Checking Correctness Of Derivations And Theory:**

I carefully checked the derivations and theory.

**Review Assessment: Checking Correctness Of Experiments:**

I carefully checked the experiments.

**Review Assessment: Thoroughness In Paper Reading:**

I read the paper thoroughly.

---

> ### Author Response · Authors · 2019-11-13
> **Some clarifications and additional ablations on discretised continuous control tasks**
>
> Thank you for your positive comments as well as careful feedback, which have led us to run some additional experiments with relevant results, shown in Appendix A.1 of the revised paper.
>
> Off-action-space: We agree that the wording of this section of the discussion is somewhat too strong as it stands, and have revised accordingly. We have now run this ablation on the toy experiments as well, and found in this case that the on-action-space version of the algorithm performed similarly to full GAS on Acrobot, and significantly worse on Mountain Car. This should help support the overall claim that the off-action-space learning is at least often valuable. Clearly the other aspects of the algorithm also contribute to the overall performance.
>
> M&M baseline: Indeed, we use an actor-critic algorithm very close to that described in the M&M paper. We will clarify this in the paper.
> It is somewhat difficult to disentangle the other differences between the approaches from the overall learning paradigm (including, for example, eps-greedy exploration vs. entropy-regularised stochastic policies). However, we felt it was an important baseline to indicate that there is notable potential in value-based and off-policy methods in this context.
> It is difficult to devise a closely corresponding Q-learning version of M&M, largely because the value of a mixture of policies is *not* equal to the mixture of the values. It is easier to consider the “Match” part of the M&M approach in the Q-learning context, and one could imagine using an L2 matching loss between upsampled-parent and child Q-values (with a weighting that is annealed as each level is adopted into the exploration policy).
> This would take the place of the adaptive intialisation scheme we adopted, and we would expect these to have broadly similar effects.
> We implemented this and ran it on the toy problems. We also ran the Sep-Q baseline that uses neither the adaptive initialisation nor the L2 matching loss. Both ablations still use our off-action-space updates.
> Our original method performed the best, although the differences are not as stark as in the StarCraft experiments. Since the value functions for these tasks are much simpler, it is probably easier to learn the new level’s Q without relying as much on the previous level’s.
>
> We would also suggest that the performance of the Q-learning algorithms with fixed action spaces should be considered important and useful baselines, as they represent standard approaches. Since our solution method using action-space hierarchies is not yet well studied, there are no other closely suited baselines (except M&M) we are aware of.
>
> We hope these additional results and clarifications will help clarify the contribution of our work.

---

### Official Review · AnonReviewer2 · 2019-10-23
**Official Blind Review #2**

**Rating:** 3

**Review:**

Based on the intuition that smaller action space should be easier to learn, the author proposes a curriculum learning approach which learns by gradually growing the action space. An agent using simpler action space can generate experiences to be used by all the agents with larger action spaces. The author presents experiments on simple domains and also on a challenging video game. In general, it is an interesting research work. I think the author can improve the paper in the following aspect.

1. Motivation. Curriculum learning is based on the idea that tasks can be arranged in a sequential manner and those tasks learned earlier can be somehow helpful for subsequent tasks. Although it is clear that small action space should be easy to learn, it is unclear why those off-policy samples can be helpful for more complex action space. Smaller action space can correspond to a completely different optimal policy. Imagine that in a tabular environment, two actions A, B are available, and the optimal action is to always take B. Then if the agent with full action space uses the experiences generated by the agent with action space {A} may get completely wrong action values. There must be some constraint of the underlying MDP. The author may provide some experiments in tabular case to illuminate the issue.

2. Relevant works. I think the author should include some discussions regarding large action spaces, since one goal of the proposed method is to handle such situation. There are several works should be discussed. For example, Deep Reinforcement Learning in Large Discrete Action Spaces, Function-valued action space for PDE control. The former handle the large discrete action space by learning an action embedding; while the latter attempts to leverage the regularity in the action space by introducing a convolutional structure for the output of the actor network and hence the proposed method can scale to arbitrary action dimensions.

**Experience Assessment:**

I have read many papers in this area.

**Review Assessment: Checking Correctness Of Derivations And Theory:**

I assessed the sensibility of the derivations and theory.

**Review Assessment: Checking Correctness Of Experiments:**

I assessed the sensibility of the experiments.

**Review Assessment: Thoroughness In Paper Reading:**

I read the paper at least twice and used my best judgement in assessing the paper.

---

> ### Author Response · Authors · 2019-11-13
> **Clarifications on motivations and large discrete action spaces**
>
> 1. The algorithm ultimately learns a value function for every action space (and while they are initialised from the previous level’s values, they are free to fit the true value function eventually). So we would not learn the wrong action values for the unrestricted action space in the limit.
> If the state-action space is simple and disjoint, as in your suggested example, then indeed a curriculum would be ineffective. We require some need for exploration in state-space facilitated by a restricted action space, or some structure in the action space, in order to see benefits from our method.
> As a tabular example: consider a gridworld where a restricted action space effectively commits to a certain direction of movement, transitioning across neighbouring states. Exploring using this action space could efficiently find a distant goal, and then refining with the full action space could determine the most optimal path to that goal.
> However, our biggest advantages should be expected outside of the tabular domain when it is important to generalise over the state and action space, since we are able to more efficiently learn useful representations from data collected using the restricted action spaces (assuming this data corresponds to more “meaningful” trajectories than the jitter induced by random exploration in a too-large action space).
>
> 2. These works are certainly somewhat relevant in that they handle large discrete action spaces, and we will include a discussion of these methods.
> However, neither explicitly addresses exploration (they key obstacle overcome by our curriculum approach). In [1], for example, the environment described in section 7.2 is explicitly designed such that fully random exploration achieves the goal reliably — in our problems and in many real-world settings, this is not the case and structured exploration is valuable. In [2], the PDE domains provide fairly dense reward signals so there is not a challenging exploration problem.
>
> These works also rely on different types of assumptions about the action space. [1] requires dense action representations given as prior knowledge in order to perform a nearest-neighbours lookup. [2] requires similar “action descriptors” in order to generalise over action dimensions. Our required hierarchy is comparatively sparse. However, if such action embeddings are available we could use them to automatically construct a hierarchy.
>
> The “adapter” of [2], used to construct the full PDE action from discretised actor outputs, is perhaps more closely related to our hierarchy since it uses regularities in the action space to connect a lower-dimensional action space to a higher (infinite) dimensional one. However, they only use this relationship to translate an action to the PDE’s space, rather than using the decomposition of the action space explicitly to shape exploration and accelerate learning as we do. We believe these types of PDE control domains would actually be strong candidates for the use of our method, if exploration was a bottleneck to learning.

---

### Official Review · AnonReviewer3 · 2019-10-24
**Official Blind Review #3**

**Rating:** 6

**Review:**

Summary: This paper proposes a method to progressively explore the action space for RL. The proposed method is called “growing action spaces”. The basic idea is that actions can usually be grouped by a hierarchical structure: the lowest level is the coarsest and higher levels gradually refine the action partition. This method effectively captures many RL settings, including multi-agent learning. One effective approach is to apply the action hierarchy.

Then the paper performs experiments on both simple toy examples and a more complicated one, the Starcraft game. The simple toy problems is “Acrobot” and “Mountain Car” with discretized spaces. A hierarchy of level 3 is considered. The experiments clearly demonstrate the effectiveness of the proposed methods. Also, the behavior of each level of the hierarchy is shown: coarse level of actions help exploration but cannot reach the highest value.

The paper then demonstrates extensive experiments on the game StarCraft. 50-100 units on each side of the game are used — which is much larger than previous papers. Further difficulties are introduced by randomizing the initializing location, and scripted logic controlled opponents. A 2-level hierarchy are tested and obtain consistently good results in all experiments. Further ablations are tested and explanations are provided.

Evaluation: Overall, I like this paper. This paper is well-written, everything is clearly explained. The proposed method is novel and effective.

Some minor comments:
	• What is the epsilon in Figure 1?
	• Using 3-level of hierarchy in the StarCraft game does not work well. You said the possible reason might be that the high level is pushing the limit of CNN. Did you try a different architecture that has a higher resolution?
	• The training curriculum is still mysterious. Why you pick a random l to start training? Maybe training level by level is better? By starting from the lowest level, you may be able to stop at any level. So you will have l different policies. In practice, you may find the best l this way.


**Experience Assessment:**

I have published in this field for several years.

**Review Assessment: Checking Correctness Of Derivations And Theory:**

I assessed the sensibility of the derivations and theory.

**Review Assessment: Checking Correctness Of Experiments:**

I assessed the sensibility of the experiments.

**Review Assessment: Thoroughness In Paper Reading:**

I read the paper thoroughly.

---

> ### Author Response · Authors · 2019-11-13
> **Thank you for positive feedback! Some answers to the minor comments:**
>
> Thank you for your positive comments and feedback!
>
> Fig 1: The epsilon (for epsilon-greedy exploration) in this experiment is annealed linearly from 1.0 to 0.1 over 25k environment steps (details of all hyperparameters in Appendix A.1). For the (slow eps) experiment the decay period is instead 100k environment steps.
>
> We did not try a higher resolution CNN; since these experiments are fairly costly to run we did not want to perform any architecture search.
>
> We have clarified the scheduling described in section 4.4. We do start from the lowest level and slowly shift to higher levels. Initially, we only use the lowest level. Then, we increase a parameter alpha which controls a Bernoulli sample between the current level and the next level (e.g. at alpha=0.1, we use the l=0 level 90% of the time, and sample the next level 10% of the time; when alpha=1.0 we use l=1 only, etc). Your suggestion of adaptively stopping the growth of the action space when learning has stopped improving is sensible, and we may try such adaptive schemes in future work.

---

### Decision · Program_Chairs · 2019-12-19

**Decision:**

Reject

**Comment:**

This paper presents a novel approach to learning in problems which have large action spaces with natural hierarchies. The proposed approach involves learning from a curriculum of increasingly larger action spaces to accelerate learning. The method is demonstrated on both small continuous action domains, as well as a Starcraft domain.

While this is indeed an interesting paper, there were two major concerns expressed by the reviewers. The first concerns the choice of baselines for comparison, and the second involves improving the discussion and intuition for why the hierarchical approach to growing action spaces will not lead to the agent missing viable solutions. The reviewers felt that neither of these were adequately addressed in the rebuttal, and as such it is to be rejected in its current form.